# In Silico Prediction of Hub Genes Involved in Diabetic Kidney and COVID-19 Related Disease by Differential Gene Expression and Interactome Analysis

**DOI:** 10.3390/genes13122412

**Published:** 2022-12-19

**Authors:** Ulises Osuna-Martinez, Katia Aviña-Padilla, Vicente Olimon-Andalon, Carla Angulo-Rojo, Alma Guadron-Llanos, Jose Carlos Rivas-Ferreira, Francisco Urrea, Loranda Calderon-Zamora

**Affiliations:** 1Biological and Chemical Sciences Faculty, Autonomous University of Sinaloa, Calzada de las Americas y calle Universitarios, s/n. Ciudad Universitaria, Culiacan 80013, Mexico; 2Department of Crop Sciences, University of Illinois at Urbana–Champaign, Urbana, IL 61801, USA; 3Centro de Investigacion y de Estudios Avanzados del I.P.N. Unidad Irapuato, Irapuato 36821, Mexico; 4Biology Faculty, Autonomous University of Sinaloa, Calzada de las Americas y calle Universitarios, s/n. Ciudad Universitaria, Culiacan 80013, Mexico; 5Medicine Faculty, Autonomous University of Sinaloa, Calle Sauces, Fraccionamiento Los Fresnos s/n, Culiacan 80019, Mexico; 6Department of Gynecology and Obstetrics, Medicine Faculty, Universidad de Concepcion, Calle Janequeo esquina Avenida Chacabuco s/n, Concepcion 4030000, Chile; 7Institute for Social Security and Services for State Workers, Regional Hospital, Heroico Colegio Militar s/n, 5 de Mayo, Culiacan 80230, Mexico

**Keywords:** in silico analysis, diabetic kidney disease, COVID-19, metabolic pathways, hub genes, potential therapeutic

## Abstract

Diabetic kidney disease (DKD) is a frequently chronic kidney pathology derived from diabetes comorbidity. This condition has irreversible damage and its risk factor increases with SARS-CoV-2 infection. The prognostic outcome for diabetic patients with COVID-19 is dismal, even with intensive medical treatment. However, there is still scarce information on critical genes involved in the pathophysiological impact of COVID-19 on DKD. Herein, we characterize differential expression gene (DEG) profiles and determine hub genes undergoing transcriptional reprogramming in both disease conditions. Out of 995 DEGs, we identified 42 shared with COVID-19 pathways. Enrichment analysis elucidated that they are significantly induced with implications for immune and inflammatory responses. By performing a protein–protein interaction (PPI) network and applying topological methods, we determine the following five hub genes: *STAT1*, *IRF7*, *ISG15*, *MX1* and *OAS1*. Then, by network deconvolution, we determine their co-expressed gene modules. Moreover, we validate the conservancy of their upregulation using the Coronascape database (DB). Finally, tissue-specific regulation of the five predictive hub genes indicates that *OAS1* and *MX1* expression levels are lower in healthy kidney tissue. Altogether, our results suggest that these genes could play an essential role in developing severe outcomes of COVID-19 in DKD patients.

## 1. Introduction

Diabetes mellitus (DM) is a chronic-degenerative endocrine disease with an insidious development [1,2]. Notably, kidney failure is a biological process closely linked to DM; above 40% of people with diabetes are likely to develop DKD [1,3]. This disease, also known as nephropathy, is a growing health condition worldwide. DKD is an outcome of hyperglycemia disrupting the glomeruli. It is defined as a progressive chronic disease, that could lead to dialysis and kidney transplant [4]. Hence, kidney failure is a life-threatening condition and patients suffering from this disease are a highly susceptible population.

Remarkably, since the beginning of the COVID-19 pandemic, it has been observed that comorbidities, including the development of diabetes chronic kidney disruptions, are associated with severe symptoms in patients with SARS-CoV-2 variants [5,6,7,8]. Notably, a higher mortality rate of COVID-19 has been reported in end-stage renal disease and kidney transplant patient recipients. Recent literature describes SARS-CoV-2 infections in diabetes and DKD; for detailed review, see [7,9]. These medical literature points out that patients with both pathologies (DKD and COVID-19) develop multiple pathophysiological mechanisms between the lung and kidney organs. For instance, the SARS-CoV-2 infection relies on angiotensin-converting enzyme 2 as an input receptor for tubular epithelial and lung epithelial cells [10,11].

Regardless of insulin treatment, diabetic patients are susceptible to developing nephropathy and approximately 50% of these patients will have end-stage renal disease. Among the processes associated with DKD are systemic and renal inflammation, where inflammatory cells such as macrophages play a crucial role, as well as genes such as proinflammatory cytokines (IL-1, IL-6 and TNF-α) and nuclear transcription factor-kappa B (NFkB). Previous reports have highlighted pathways such as Janus kinase/signal transducer and the activator of transcription (JAK/STAT) as pivotal for developing this disease [12]. Moreover, recent studies have determined that IL-1β, IL-6 and TNF-α cytokines participate in the cytokine storm in patients with COVID-19, producing renal inflammation and cardiovascular disorders as an outcome.

In 2020, Wu et al. reported the clinical manifestations of 49 hospitalized hemodialysis patients infected with COVID-19 who developed pneumonia. Overall, the disease course was severe in patients with kidney failure. Notably, patients undergoing hemodialysis with COVID-19 were at higher risk of death [13]. Other studies suggest that, among the complications that occur in hemodialysis, patients with non-invasive ventilation develop acute respiratory distress syndrome (ARDS), shock, acute cardiac injury and arrhythmia, which were less frequent conditions in patients without dialysis [7,14]. These observations suggest that atypical clinical presentations of COVID-19 developing in hemodialysis patients should be considered. Hence, there is a current need to implement a systematic approach to diagnose atypical cases of COVID-19. For instance, atypical symptoms, such as fatigue, diarrhea and loss of appetite are also present in other diseases, complicating its diagnosis.

Moreover, Mourad et al. reported that viral pneumonia occurred in 96% of hospitalized patients, of which 39% received mechanical ventilation and 21% received renal replacement therapy. Notably, 28% of kidney transplant patients and 64% of intubated patients died. In this context, building guidelines for managing and considering atypical signs and symptoms for differential diagnosis in kidney transplant and dialysis patients are needed to reduce complications during the procedure. In keeping with this, Kataria et al. reported management considerations for patients with COVID-19 who need a kidney transplant to reduce the spread of the virus in donors, recipients and medical staff [7,15].

Even with the development of vaccines and their worldwide distribution, it is noteworthy that we must consider the constant interaction with new and possibly more infectious variants [14,16,17]. In this sense, the continuing global spread of SARS-CoV-2 and the latent morbidity/mortality of COVID-19 increases with comorbidities, such as DKD. Hence, it is necessary to develop strategies to improve patients’ health.

In this context, bioinformatics approaches are powerful strategies, considered one of the most promising tools to predict potential therapeutic genes with clinical relevance and the biological processes underlying complex diseases [18,19]. Recent studies have identified predictive genes shared among COVID-19 and multiple comorbidities by performing bioinformatics analysis. These approaches have provided insights into molecular mechanisms and the biological processes, in order to divert the possible result of a severe COVID-19 condition and to provide potential therapeutic targets [20,21,22].

In this work, we focus on transcriptomics and network analysis to determine the shared genes between DKD and COVID-19, emphasizing the identification of those which represent hub genes involved in kidney disease complications. Our study could contribute to a better understanding of the pathophysiological impact of COVID-19 on kidney disease complications in diabetic patients. We consider that this could serve as a basis for further research in developing improved therapy for patients suffering from this disease.

## 2. Materials and Methods

For this study, we designed a bioinformatics pipeline that uses transcriptomic data to identify predictive hub genes involved in DKD complications derived from COVID-19 infection as depicted in Figure 1. Used codes are available at https://github.com/kap8416/Transcriptomics-Diabetes-Kidney-Disease, accessed on 7 October 2022.

### 2.1. Transcriptomic Data Acquisition

The microarray expression dataset (GSE30529) was obtained from the NCBI GEO database (https://www.ncbi.nlm.nih.gov/geo/, 22 June 2022). This dataset is based on the GPL571 platform and comprises human samples from patients with DKD (*n* = 10), whose glomerular filtration rate (eGFR) average is 21.85 ± 11.54 and healthy (non-diabetic) controls (*n* = 12) is 73.77 ± 21.08.

### 2.2. Differential Gene Expression Analysis

Raw data was processed with the *affy* package available in Bioconductor using the robust multi-array average (RMA) function to convert an *AffyBatch* object into an *ExpressionSet* object (https://www.bioconductor.org/, 22 June 2022). Then, data normalization was performed. After that, we removed the *batch effect* using *ComBat* from the *sva* and applied Weighted Gene Co-Expression Network Analysis (WGCNA) to identify outliers. The *collapseRows* function, available in the WGCNA package for collapsing gene expression of several probes in a single measurement per gene, was used. Finally, differential gene expression analysis was performed using the limma package. Log2FC > 1 and FDR < 0.05 values were considered to determine the differentially expressed genes (DEGs). Genes having significant *p*-values with positive Log2FC represent an increased expression (UP). Those with negative Log2FC values are considered downregulated (DN), while those with *p*-values above 0.05 do not have changes between stages (NC).

### 2.3. Functional Enrichment Analysis for DEGs in the DKD Dataset

Functional enrichment analysis was assessed for the 995 DEGs in the DKD dataset using the Gene Set Enrichment Analysis (GSEA) of DEGs. The enrichment analysis includes all the Gene Ontology (GO) terms, the Kyoto Encyclopedia of Genes and Genomes (KEGG) and Reactome pathways. The cluster profiler R package [23] was employed for data analysis and SRplot server (http://www.bioinformatics.com.cn/en?keywords=pie, accessed on 10 August 2022) for data visualization.

### 2.4. Functional Enrichment Analysis for the Shared Pathogenic DEGs

Functional enrichment analysis was performed for the 42 shared DEGs using the GO terms for biological processes, KEGG and Reactome pathways. The Cluster profiler R package was employed for data analysis and visualization [23].

Then, the ClueGO (version 2.5.7) module of the Cytoscape software (version 3.9.0) was used to examine the inter-relational pathways of the significantly enriched functions to identify the most significant genes [24]. We identified up and downregulated pathways using the top 200 DEGs. Then, the 42 overlapped genes were considered to perform the same analysis with a kappa score of 0.4, *p*-value < 0.05. For statistical tests, hypergeometric two-sided and Benjamini–Hochberg methods were employed [24].

### 2.5. PPI Network and Identification of Hub Genes

The list of 41 up-DEGs previously identified in the shared pathogenetic process were analyzed for deep insight into their interatomic roles. Network analysis and visualization were performed using the Cytoscape version 3.9.0. STRINGDB platform (https://string-db.org/, accessed on 14 September 2022) was used to obtain physical and functional experimental validated data. For this study, high confidence scores (0.7) and no additional interactors were filtered, keeping only experimental, co-expression and database characterized interactions. Each node in the PPI network represents a protein, whereas edges are interactions and connections between them. Finally, the predictive hub genes were obtained using the CytoHubba module selecting the top five nodes, employing the topological algorithm of Maximal Clique Centrality (MCC) [25].

### 2.6. Deconvolution of the Gene Regulatory Network (GRN)

In order to identify co-expression modules of the five hub genes, the *Corto* algorithm with default parameters was used for the GRN inference. This package is freely available on the CRAN repository. The *Corto* algorithm is implemented as a co-expression-based tool that infers GRNs using as an input a TFs list with their targets and an expression matrix data set [26]. In brief, *Corto* uses a combination of Spearman correlation and Data Processing Inequality (DPI), adding bootstrapping to evaluate the significant edges by removing indirect interactions. We used the DKD expression matrix and a human-TF list as input. For this analysis, a *p*-value = 1 × 10^−8^ was used as a cut-off and 100 bootstraps were carried out. Subsequently, an output file containing an inferred enriched GRN was obtained. For network visualization, the Cytoscape environment was used to identify the co-expressed gene modules of each hub gene [27].

### 2.7. Functional Enrichment Analysis for Co-Expressed DEGs in the GRN

Functional enrichment analysis was assessed using the GSEA of each of the five previously identified modules of the hub genes’ co-expressed DEGs. Functional enrichment considered all the GO terms and the KEGG and Reactome pathways. The cluster profiler R package was employed for data analysis [23]. For data visualization the parental terms were depicted in the GRN using the Biorender environment (https://biorender.com/, accessed on 2 September 2022).

### 2.8. In Silico Validation of Expression of the Predictive Hub Genes in Coronascape DB

Gene expression data was accessed at the Coronascape repository https://metascape.org/COVID/, accessed on 14 September 2022 [28]. Coronascape is a resource for the analysis of systems-level datasets. The top statistically overlapped COVID reference lists datasets were analyzed against our gene list (5 hub-DEGs). DEGs conserved among the COVID databases were identified and visualized in Circle Plot using the Circos visualization tool (https://genome.cshlp.org/content/early/2009/06/15/gr.092759.109.abstract, accessed on 14 September 2022) for comparative genomics analysis.

### 2.9. In Silico Validation of Expression of the Predictive Hub Genes in GTEx Database

Gene expression data was accessed at the GTEx repository (https://www.gtexportal.org/, accessed on 16 September 2022). GTEx is a resource for the analysis of tissue-specific level datasets. The five hub genes were tested for multi-gene query expression in kidney, lung, heart, pancreas and whole blood healthy tissues.

## 3. Results

### 3.1. Identification of Differentially Expressed Genes across Diabetic Kidney Disease

First, we performed raw data normalization using the No GSE305029 Affymetrix microarray dataset. This step allowed reducing false positives, as depicted in Figure 2a. Later, principal component analysis (PCA) showed clustered distances among each renal tubule sample, including those belonging to the disease condition (DKD) and those to the control group (Figure 2b). After this step, one batch was observed for the sample GSM757034 belonging to the control group. This sample was removed for further dataset processing. After that, we performed WGCNA. This approach is a widely used data mining and exploratory method. It allowed us to identify clusters and outliers to discard samples that did not cluster with their corresponding group (disease or control). Following this step, we removed sample GSM757027. After that, two separate clusters according to their patient phenotype are depicted in Figure 2c and employed further for the transcriptomics analysis. By performing gene expression analysis, we identified a total of 13,309 genes in the microarray samples. Nine hundred ninety-five out of 13,309 genes are DEG in the disease samples when compared to kidney tubules. Remarkably, our results showed that DEGs are significantly upregulated. Around ~63% of them (632 genes) are induced, while only 363 were undergoing down-regulation of their expression (Figure 2d, Appendix A).

The top ten upregulated genes identified are *IGHA2* (Log2FC = 6.027, FDR = 9.88 × 10^−11^), which encodes immunoglobulin receptor binding. This gene is involved in glomerular filtration and its related pathway is the inflammatory response pathway. Followed by *IGHGP* (Log2FC = 5.749, FDR = 8.16 × 10^−9^), a pseudogene accordingly non-functional gene. Then, *LTF* (Log2FC = 5.482, FDR = 4.32 × 10^−13^) is a member of the transferrin family of genes with antimicrobial, antifungal, antiparasitic and antiviral activity against both DNA and RNA viruses, including activity against SARS-CoV-2. Meanwhile, *IGLC2* (Log2FC = 5.320, FDR = 3.27 × 10^−9^) is a gene involved in several processes, including activation of the immune and defense responses to other organisms and phagocytosis. This gene is involved in glomerular filtration and its related pathway is the inflammatory response pathway. Followed by *IGHGP* (Log2FC = 5.749, FDR = 8.16 × 10^−9^) a pseudogene accordingly non-functional gene. Then, *LTF* (Log2FC = 5.482, FDR = 4.32 × 10^−13^) is a member of the transferrin family of genes with antimicrobial, antifungal, antiparasitic and antiviral activity against both DNA and RNA viruses, including activity against SARS-CoV-2. Meanwhile, *IGLC2* (Log2FC = 5.320, FDR = 3.27 × 10^−9^) is a gene involved in several processes, including activation of the immune and defense responses to other organisms and phagocytosis. Additionally, it is related to the network map of the SARS-CoV-2 signaling pathway, followed by *JCHAIN* (Log2FC = 4.861, FDR = 1.17 × 10^−11^). This gene participates in IgA binding activity and protein homodimerization activity and contributes to several processes, including glomerular filtration and positive regulation of respiratory burst. *CXCL6* (Log2FC = 4.793, FDR = 3.73 × 10^−12^) have an effect against gram-positive and gram-negative bacteria. *LYZ* (Log2FC = 4.710, FDR = 1.02 × 10^−10^) is a gene encoding a protein with antimicrobial activity in human milk and other tissues such as lungs and kidneys. *C3* (Log2FC = 4.587, FDR = 3.53 × 10^−11^) is a protein-coding gene that plays a central role in the activation of the complement system. This gene encoded a peptide that modulates inflammation and possesses antimicrobial activity. *IGHM* (Log2FC = 4.390, FDR = 2.60 × 10^−7^) is a gene that encodes the C region of the heavy mu chain, which defines the IgM isotype. *ALOX5* (Log2FC = 4.261, FDR = 6.81 × 10^−10^) is a lipoxygenase gene that plays a dual role in the synthesis of leukotrienes from arachidonic acid. Notably, leukotrienes are essential mediators of several inflammatory and allergic conditions. (Figure 3, Appendix A).

Opposite, the top ten downregulated DEGs listed are the following. *CYP27B1* (Log2FC = −3.693, FDR = 1.24 × 10^−6^) encodes a protein member of the cytochrome P450 superfamily of enzymes and is related to the steroid metabolism pathways. Followed by *HRG* (Log2FC = −3.661, FDR = 1.09 × 10^−8^), a gene encoding a histidine-rich glycoprotein located in plasma and platelets, but the physiological function has yet to be determined. However, the encoded peptide displays antimicrobial activity against *Candida albicans, Escherichia coli, Staphylococcus aureus, Pseudomona aeruginosa and Enterococcus faecalis* microorganisms [29]. Followed by *G6PC1* (Log2FC = −3.575, FDR = 1.87 × 10^−7^), a gene that catalytic-subunit-encoding glucose-6-phosphatase enzyme is involved in glucose homeostasis. *KNG1* (Log2FC = −3.383, FDR = 1.76 × 10^−6^) encodes high molecular weight kininogen (*HMWK*) and low molecular weight kininogen (*LMWK*) proteins. *HMWK* is essential for blood coagulation and the release of bradykinin [30,31]. Interestingly, during SARS-CoV-2 infection, an increase in bradykinin levels is associated with lung injury and inflammation [32]. Meanwhile, *ALB* (Log2FC = −3.188, FDR = 1 × 10^−3^) encodes the most abundant protein in human blood. This protein regulates blood plasma colloid osmotic pressure and acts as a carrier protein for many endogenous genes. Additionally, this protein encodes a peptide, an endogenous inhibitor of the *CXCR4* chemokine receptor. *APOH* (Log2FC = −3.067, FDR = 1.09 × 10^−5^) is a gene coding by apolipoprotein H, a component of circulating plasma lipoproteins. *EGF* (Log2FC = −3.017, FDR = 8.05 × 10^−8^) this gene encodes a member of the epidermal growth factor superfamily. This protein is a potent mitogenic factor that plays an important role in numerous growth, proliferation and differentiation cell types. *GPC5* (Log2FC = −2.746, FDR = 4.63 × 10^−6^), a protein-coding gene related to the SARS-CoV-2 infection pathway, is also identified as downregulated. Besides, *PLPPR1* (Log2FC = −2.711, FDR = 5 × 10^−5^), a gene of member plasticity-related gene (*PRG*) family and *UMOD* (Log2FC = −1.61 × 10^−8^, FDR = 5.64 × 10^−7^) which is the most abundant protein in mammalian urine under physiological conditions and its urine excretion may defend against urinary tract infections caused by uro-pathogenic bacteria [33,34].

Altogether, our differential expression analysis results show that the most upregulated genes are a group of immunoglobulin receptors that could activate the immune and defense responses to other organisms and phagocytosis. Meanwhile, the downregulated genes are associated with mitochondria and cell compartment metabolism, particularly lipoproteins.

### 3.2. DEGs in DKD Are Significantly Enriched in Cell and Immune Responses and COVID-19 Affected Pathways

To delve into the roles of the DEGs, we performed a functional enrichment analysis (Figure 4). Our results indicates that the DEGs are highly enriched in the following immunological biological processes: leukocyte cell-cell adhesion (count = 65, *p* = 5.10 × 10^−28^), leukocyte proliferation (count = 62, *p* = 4.78 × 10^−29^), adaptive immune response based on somatic recombination of immune receptors built from immunoglobulin superfamily domains (count = 61, *p* = 3.48 × 10^−25^), positive regulation of cell activation (count = 60, *p* = 1.52 × 10^−22^), regulation of leukocyte cell-cell adhesion (count = 59, *p* = 4.09 × 10^−25^), extracellular matrix organization (count = 59, *p* = 1.97 × 10^−24^), extracellular structure organization (count = 59, *p* = 2.22 × 10^−24^), positive regulation of leukocyte activation (count = 59, *p* = 7.37 × 10^−23^), regulation of T cell activation (count = 57, *p* = 1.78 × 10^−23^), humoral immune response (count = 57, *p* = 1.95 × 10^−22^), regulation of leukocyte proliferation (count = 55, *p* = 1.10 × 10^−27^), lymphocyte proliferation (count = 55, *p* = 5 × 10^−25^), mononuclear cell proliferation (count = 53, *p* = 7.48 × 10^−25^) and positive regulation of lymphocyte activation (count = 53, *p* = 9.44 × 10^−21^) (Figure 4a, Appendix A).

Among those, phagosome (count = 40, *p* = 4.57 × 10^−22^), allograft rejection (count = 18, *p* = 1.10 × 10^−15^), Epstein-Barr virus infection (count = 37, *p* = 6.80 × 10^−15^), viral myocarditis (count = 21, *p* = 7.51 × 10^−15^), graft-versus-host disease (count = 18, *p* = 1 × 10^−14^), type I diabetes mellitus (count = 18, *p* = 1.66 × 10^−14^), *Staphylococcus aureus* infection (count = 25, *p* = 4.67 × 10^−14^), Coronavirus disease-COVID-19 (count = 30, *p* = 1.82 × 10^−13^), Leishmaniasis (count = 22, *p* = 2.03 × 10^−13^), cell adhesion biomolecules (count = 30, *p* = 9 × 10^−13^), tuberculosis (count = 32, *p* = 1.20 × 10^−12^), autoimmune thyroid disease (count = 18, *p* = 1.21 × 10^−12^), rheumatoid arthritis (count = 23, *p* = 1.63 × 10^−12^) and antigen processing and presentation (count = 20, *p* = 2.36 × 10^−11^) are induced pathways (Appendix A).

In contrast, small molecule catabolic process (count = 65, *p* = 9.17 × 10^−12^), organic acid catabolic process (count = 53, *p* = 1.07 × 10^−39^) and carboxylic acid catabolic process (count = 53, *p* = 1.07 × 10^−39^) are repressed biological processes. Likely (Figure 3B, Appendix A) route blades, butanoate metabolism (count = 10, *p* = 1.32 × 10^−9^), valine, leucine and isoleucine degradation (count = 12, *p* = 4.83 × 10^−9^) and peroxisome related pathways (count = 14, *p* = 4.81 × 10^−8^) (Figure 4b, Appendix A).

Interestingly, upregulated DEGs were implicated in autoimmune disorders and bacterial and viral infections such as COVID-19, whereas downregulated DEGs were involved with amino acid metabolism pathways. Altogether, the most enriched pathways demonstrate the significant role of the DEGs in diabetic nephropathy, activating immunological and defense responses commonly affected in the SARS-CoV-2 scenario. Opposite, the top enriched biological processes are linked to amino acid and metabolism pathways, suggesting their role in the consequence is a profound disturbance in glycolysis and lipid and amino acid metabolism.

### 3.3. Transcriptional Reprogramming of Gene Upregulation Is a Common Mechanism for DKD and COVID-19 Conditions

Notably, our results showed that ~67.6% of DEGs in DKD samples are upregulated. From the 632 upregulated genes, 41 out of the 560 genes are involved in COVID-19 pathways (Figure 5a, Appendix A). Interestingly, the overlapped DEGs among DKD and COVID-19 are affecting pathways that include immune, diabetes and other comorbidities according to their role in KEGG, Reactome and GO terms (Figure 5b).

Among the top 15 genes involved in the identified pathways are those described in Table 1. *C3* is highly significant, this gene plays a central role in the activation of the complement system and encodes the C3a peptide, which modulates inflammation and possesses antimicrobial activity [35], while *CFB* gene is a component of the alternative pathway of complement activation, which circulates in the blood as a single-chain polypeptide. *CASP1* plays a central role in cell apoptosis. This gene activates the inactive precursor of interleukin-1, a cytokine involved in the processes such as inflammation. *TLR7* and *TLR2* are essential in recognizing pathogen-associated molecular patterns (PAMPs) by infectious agents that mediate the production of cytokines necessary for developing effective immunity. *TLR7* participates in recognition of single-stranded RNA viruses, which this gene is associated with COVID-19 [36]. *CXCL10*, *CXCL8* and *CCL2* are cytokine genes involved in inflammatory processes. *CXCL10* is a gene key regulator of the cytokine storm immune response to SARS-CoV-2 infection and *CCL2* is associated with severe acute respiratory syndrome coronavirus 2 [37,38,39].

Moreover, *CXCL8* plays a role in the pathogenesis of the lower respiratory tract infection bronchiolitis, a common respiratory tract disease caused by the respiratory syncytial virus (RSV). In comparison, *JAK1* is a member of a class of protein-tyrosine kinase. This gene is a component of the *IL6/JAK1/STAT3* immune and inflammation response and is a therapeutic target for alleviating cytokine storms. *MX1* encodes a guanosine triphosphate, a metabolizing protein that participates in the cellular antiviral response. This protein is induced by type I interferon like *IFNAR2* and antagonizes the replication process of several different RNA and DNA viruses. Another identified gene, *IRF7*, encodes interferon regulatory factor 7, a member of the interferon regulatory transcription factor (IRF) family. This gene plays a role in the transcriptional activation of virus-inducible cellular genes and participates in the innate immune response against DNA and RNA viruses. While *STAT1* is a gene that can be activated by different ligands, including interferon-α, interferon-γ, *EGF*, platelet-derived growth factor and *IL6*. This protein mediates the expression of various genes and plays an essential role in immune responses to viral and other pathogens. Consequently, *ISG15* is a ubiquitin-like protein conjugated to intracellular target proteins upon activation by interferon-α and interferon-β-like *STAT1* activates interferon-α. *OAS1* plays a crucial role in the innate cellular antiviral response, including SARS-CoV-2 [34,40].

Then, functional enrichment analysis was performed to determine the predictive roles of the 41 common upregulated genes. As could be expected, in accordance with the enrichment analysis of the total upregulated DEGs, the most enriched pathways are those associated with defense and immune responses (Figure 5c). For instance, complement activation, defense response and humoral immune response were the top enriched terms. Moreover, other affected processes identified in the upregulated genes are clustered in angiogenesis and vasculature development events. Besides, these induced genes participate in cellular response to viruses and to cytokine stimulus.

Further, to determine their involvement in multiple molecular-biological processes, an inter-relational analysis was performed using the ClueGO module of the Cytoscape environment to obtain functionally organized pathway-terms networks. ClueGO employs kappa statistics to link the functional terms in the network. As shown in Figure 6a, the upregulated DEGs were mainly involved in the innate immune system, leukocyte differentiation, angiogenesis regulation, cytokine induction, extracellular matrix organization, defense response to other organisms and regulation of peptidase activity. Meanwhile, the inter-relational analysis of the downregulated genes reveals that the most influenced connected pathways are associated with fatty acid and lipid metabolic processes. However, other essential routes are highlighted in this analysis even though they are not highly associated with each other—for instance, anion transport, response to insulin and renal system development (Figure 6b). Markedly, overlapping genes were inter-relational mainly in routes in SARS-CoV infections, AGE-RAGE signaling, Interferon signaling and complement activation (Figure 6c).

To gain insight into the genes participating in the inter-relational pathways, we identify the potential biological involvement of the upregulated DEGs in KEGG, Reactome and GO metabolic pathways (Figure 7). Notably, our results show that the five enriched terms in KEGG are coronavirus disease-COVID-19 (count = 30, *p* = 4.95 × 10^−38^), followed by complement and coagulation cascades (count = 10, *p* = 1.10 × 10^−11^), type I diabetes mellitus (count = 5, *p* = 2.35 × 10^−6^), lipid and atherosclerosis (count = 8, *p* = 1.11 × 10^−5^) and AGE-RAGE signaling pathway in diabetic complications (count = 5, *p* = 1.48 × 10^−4^).

We identified that *C3*, *C1QA*, *C1QB*, *C1S*, *CFB*, *CXCL10* and *VWF* were associated with coronavirus disease and complement and coagulation cascades, while *CYBB, STAT1* and *PRKCB* were involved in coronavirus disease COVID-19 and AGE-RAGE signaling pathway in diabetic complications. Coronavirus disease-COVID-19 and lipid and atherosclerosis connect with *TRAF3*, *TLR2*, *SELP* and *CASP1* deregulated molecules. Finally, *CCL2* and *CXCL8* genes are related in three pathways (Figure 7a). Meanwhile, for the Reactome pathways analysis, the five enriched pathways are regulation of complement cascade (count = 8, *p* = 9.55 × 10^−12^), *ISG15* antiviral mechanism (count = 5, *p* = 9.44 × 10^−6^), diseases of the immune system (count = 3, *p* = 1.15 × 10^−4^), SARS-CoV infections (count = 4, *p* = 2.78 × 10^−3^) and potential therapeutics for SARS (count = 3, *p* = 4.14 × 10^−3^). IFNAR2, TLR7 and JAK1 genes connect SARS-CoV infections and potential therapeutics for SARS. Moreover, *JAK1* gene is also present in the *ISG15* antiviral mechanisms (Figure 7b). Meanwhile, the interactions within the biological process are regulation of immune effector process (count = 16, *p* = 1.18 × 10^−12^), positive regulation of cytokine production (count = 13, *p* = 1.20 × 10^−11^), regulation of complement activation (count = 8, *p* = 1.81 × 10^−10^), defense response to virus (count = 10, *p* = 2.48 × 10^−10^) and toll-like receptor signaling pathway (count = 6, *p* = 1.18 × 10^−6^). Hence, *C1QA*, *C1QB*, *C1S*, *C1R*, *CFB* and *C7* regulate the immune effector process and complement activation. *C3* and *C3AR1* are also participating in the positive regulation of cytokine production, while *HLA-A*, *HLA-E*, *HLA-F*, *HLA-G* and *SKY* regulate the immune effector process and positive regulation of cytokine production. Moreover, *TLR1* and *TLR2* are involved with the positive regulation of cytokine production and toll-like receptor signaling pathway, and *IRF7*, *TLR7*, *TRAF3* and *STAT1* are present in more biological processes (Figure 7c).

Our results show that ~67.6% of DEGs in DKD samples are upregulated. Around 97% (41/42) of the DEGs associated with COVID-19 pathways are upregulated. Those genes participate in crucial biological and metabolic processes, including complement and coagulation cascades, lipid and atherosclerosis, AGE-RAGE signaling pathway and positive regulation of cytokine production. Notably, according to the Reactome database, we characterize some of those induced genes as potential therapeutic targets for SARS-CoV-2 infection. In summary, our transcriptomic and functional enrichment analysis suggests that transcriptional reprogramming of gene upregulation is a common mechanism in DKD and COVID-19 conditions linked to potential pathways involved in the development of other comorbidities.

### 3.4. Interactome Analysis and Hub Genes Identification

The PPI network analysis is a useful method for identifying the hub genes playing a critical role in complex disease interactomes. The PPI networks for up-DEGs are shown in Figure 8a. In the predicted PPI networks for the 41 up-DEGs, an operation with a network scoring cutoff of 0.7 was applied using the MCODE plug-in of the Cytoscape environment. It resulted in seventeen DEGs with the highest interactions in the network. Among those 17 most relevant genes, we find the following *TLR2*, *TLR7*, *CXCL10*, *TRAF2*, *MX2*, *JAK1*, *IGNAR2* and a cluster of HLA family proteins. Then, the five prominent upregulated hub genes, namely *STAT1*, *MX1*, *ISG15*, *IRF7* and *OAS1*, were identified and exhibited protein-protein-validated experimental interactions with a high confidence score of 0.7.

When analyzing the interactome of DEGs from the GRN and their regulons, a total of 12,131 nodes and 160,110 interactions were obtained (Figure 8b). When comparing DKD samples against healthy tissue, our results showed that the *STAT1* gene module has 117 interactors, followed by the *IRF7* gene module with 43 interactions, *ISG15* presents 34 co-expressed genes, *OAS1* has 12 genes and, in the *MX1* gene module, five genes are co-expressed. Functional enrichment analysis determined that the *STAT1* gene module is associated with protein localization, targeting and transport, while *IRF7* co-expressed genes participate in signaling by interleukins and negative regulation of immune response. *ISG15* is related to cardiac muscle, while the *OAS1* module participates in muscle cell proliferation and positive regulation of transcription and *MX1* co-expressed genes do not present specific functional enrichment terms (Appendix A).

Overall, our interactome analysis results show that the five hub genes identified are co-expressed in modules with other genes participating in the events associated with heart disease, virus immune response and genetic regulation at the transcriptional and post-transcriptional levels.

### 3.5. In-Silico Validation of Hub Genes across COVID-19 and Healthy Tissue Datasets

Finally, we performed in silico validation analysis to determine the expression profiles of the hub genes obtained by the interactome analysis (Figure 9). First, we employed the Coronascape Database, which represents a compendium of large-scale studies of transcriptomics data that allows us to perform biological systems analysis. When comparing the expression of our hub genes in DKD samples against this compendium, we identified that studies that comprise the following immune cell samples, dendritic, Calu3, CD4T, CD16/CD14 monocytes, NK and A549lowMOI cells, are the ones that overlapped in the expression—*ISG15* being the most conserved differentially expressed gene (8), followed by *STAT1* (7), *OAS1* (6) and *IRF2* (2) (Figure 9a).

Then, we delve into the expression of those genes in healthy tissue linked to the potential comorbidities’ development such as the pancreas, heart, kidney and lung (Figure 9b). Additionally, we included whole blood for comparison of the baseline expression levels of those genes. Our results indicate that *MX1*, *IRF7*, *ISG15* and *STAT1* possess lung tissue-specific expression. Single-cell expression testing identified that this specific expression its higher in macrophages and B and T cells (Figure 9c). In contrast, *IFR7* is a gene with the highest expression values in whole blood. While *OAS1* and *MX1* are the genes with the lower expression levels in pancreas, heart and kidney healthy samples (Figure 9b).

Hence, the in-silico validation of our hub genes in those expression datasets pointed out that *ISG15* and *MX1* are the most conserved up-regulated genes across COVID-19 datasets of immunological cells, while *OAS1* and *MX1* are those with lower gene expression levels in tissue-specific healthy samples.

## 4. Discussion

DM has been recognized worldwide as one of the most critical metabolic diseases because of the size of the affected population. Notably, more than 40% of diabetic patients are likely to develop DKD [1,3]. Those patients are at an even higher risk than patients with diabetes for severe COVID-19 disease progression [14]. This fact has highlighted the impact of DKD’s fatal outcome associated with COVID-19 infection [41]. However, there is scarce information about the biomolecules and the biological and metabolic processes involved in worsening the clinical condition triggered by COVID-19 in patients with DKD. Hence, there is an urgent need to further our knowledge of deregulated genes as it will help to design better therapies against those complications.

Herein, we characterize a particular group of upregulated genes at the transcriptional and post-transcriptional levels, including the pathogenetic events shared for DKD and COVID-19 conditions (Figure 10). Remarkably, a group of these genes is linked to biological behavior as complement and coagulation biomolecules (*C3*, *C1QB*, *CFB*, *CFD*, *C7*, *C3AR1* and *VWF*). Previous studies showed that the pathophysiological mechanism of COVID-19 is associated with the complement system [42] and the coagulation system [43]. This inter-relation could be feasible to contribute to the worsening of diabetic patients with DKD infected with the SARS-CoV-2 virus. This phenomenon has been reflected in ICU rooms, giving worse results in days of hospitalization or survival of those patients [44,45,46]. 

Interestingly, we identified the upregulation of the *CXCL10* gene. This biomolecule has been described as a molecular marker of severe conditions in autoimmune diseases and COVID-19. It is a genomic marker of poor prognosis or progression of the disease [47,48]. This molecule should be considered a possible therapeutic target if its presence is confirmed in patients with DKD. Remarkably, other DEGs upregulated in the analyzed samples of diabetic patients are strongly related to pro-inflammatory cytokines and cell activation—for instance, *PKC*, *CCL2*, *IL-8*, *SYK*, *JAK1*, *IFNAR* and *STAT1* and anti-inflammatory such as *IL-10*. As well as adhesion (*SELP*); apoptosis (*CASP1*), production of reactive oxygen intermediates (*NOX2*) and Pathogen Recognition Receptors (*PRR*: *TLR-2/4*, *TLR-7/8*). These biomolecules have been described as extremely important in the defense against the virus, particularly TLR7/8 [49]. This finding highlights the possibility of performing research testing the potential clinical relevance of TLRs as possible therapeutic targets. 

We also identified DEGs involved in chemotaxis and adhesion biomolecules, such as *MCP-1*, *IL-8* and *SELP*. These biomolecules participate in the recruitment of immune cells, mainly effectors, such as neutrophils or monocyte/macrophage type. Despite their crucial role (both are very important for host protection), if they reach the parenchyma (- in the context of overactivation described for COVID-19 condition), these cells could degranulate and generate cause more damage to target organs [50,51].

In our study, we also focused on analyzing the potential interactions of those deregulated molecules at their post-transcriptional level by performing PPI and co-expression GRN analysis. We identified 17 genes as the highest interactors among the DEGs shared in both disease conditions. Interestingly, some of the closest interactors are genes previously proposed as potential therapeutics for SARS-CoV-2. For instance, *IFNAR2*, *IMPDH2*, *ITGA4*, *JAK1*, *TLR7* and *TUBB* have been previously identified by computational predictions and annotated in the Reactome database. Strikingly, in our computation analysis, we determined that, among those gene interactors, the following five are classified as hub genes: *STAT1*, *IRF7*, *ISG15*, *MX1* and *OAS1*. Hence, our hub genes are new molecules with promising clinical potential. Notably, those hub genes are linked to pivotal biological and metabolic pathways restricted to DKD and COVID-19 conditions and include heart disease, angiogenesis disruptions and other metabolic disorders. It is noteworthy that our predictive molecules could have therapeutic potential as targets. For example, according to a proteome-wide genetic colocalization study performed by Anisul et al., *OAS1* was identified as a protein associated with the risk of COVID-19 [52]. The measured mRNA levels of *OAS1* were associated with reduced numbers of susceptibility, hospitalization, ventilation and death [53]. This study highlights that pharmacological agents that increase *OAS1* levels could be prioritized for drug development. Therefore, this evidence suggested that the induction of genes such as *OAS1* in severe COVID-19 conditions could have a better prognosis in those patients [53]. Opposite, other studies showed that *OAS1* might play a critical role in regulating the development of chronic kidney disease (CKD) [54].

Moreover, it has been reported that the over-expression of IFN-stimulated cytokine genes such as *OAS1* and *ISG15* could contribute to systemic inflammation and, as an outcome, to CKD [55]. In accordance, the Gene-Disease Associations dataset describes that, during acute kidney injury, the herein-identified five hub genes are induced [56]. This suggests their possible involvement in leading CKD.

Another hub gene interferon-induced is *MX1*. It has been reported that this molecule is highly expressed during renal diseases such as lupus nephritis and could cause kidney inflammation [57]. Moreover, another study supports this gene behavior. When analyzing its expression on hamster organotypic kidney cultures infected with SARS-CoV-2, it increased thirteenfold the level of its baseline expression compared to the controls [58]. In this context, other genes of the IFN system that could be stimulating the signaling pathway cascades are *IRF7* and *STAT1*. Notably, the STAT1 signaling pathway is associated with renal inflammation and injury [59]. An upregulation of *STAT1* has also been reported in mildly to severely affected COVID-19 patients [60].

Remarkably, we observed that DKD activates several mediators in the immune response that may be participating in the excessive production of pro-inflammatory cytokines, and causes similar cytokine storms observed in COVID-19 complications [61]. Besides, conditional profiling expression in persistent cytokine in DKD may increase susceptibility to RNA viral infections such as SARS-CoV-2 [62]. 

Overall, our interactome analysis results show that the five hub genes identified are co-expressed in modules with other genes participating in molecular events associated with heart disease, virus immune response and genetic regulation at the transcriptional and post-transcriptional levels. Our co-expression analysis highlights the regulatory role of *STAT1* in protein localization, targeting and transport. *OAS1* and *ISG15* are upregulated molecules associated with genes linked to cardiac muscle proliferation co-expression. Our interactomics approach identified the possible biological effects of severe SARS-CoV-2 infection under the DKD condition context, finding infections occurring in other organs in the body. Notably, revealing linkages with cardiovascular diseases, such approaches could represent critical steps toward new treatments.

To validate the potential of those molecules as clinic biomarkers, we performed an in silico analysis, taking advantage of publicly available databases. At the systems level, we compared the hub genes expression against datasets in the Coronascape DB. *ISG15*, *STAT1* and *MX1* are the most conserved up-regulated genes across COVID-19 datasets of immunological cells. Among immune cell samples identified to possess induction of these biomolecules in COVID-19 patients are dendritic, CD4+T, Natural killer (NK) and CD14+CD16+ monocyte, as in the lung cancer cell line (calu3 and A539). Meanwhile, *OAS1* is induced in all the immune cells described except for the CD16+ monocytes.

In contrast, the overexpression of *IRF7* was found only for the cancer cell line A539 and in dendritic cells. NK and dendritic cells are essential in responding to viral infections, but during CKD and SARS-CoV-2 infections these immune cells decrease their circulating levels [63,64]. On the contrary, activation of the intermediate (CD14−CD16+) monocytes is considered an inflammatory mediator to increase chronic conditions such as cardiovascular disease, diabetes and DKD [65,66]. It has been reported that circulating monocytes in COVID-19 had high levels in the acute and post-acute state until up to fifteen months. Besides, CKD is associated with higher pro-inflammatory serum cytokines levels and CD4+T lymphocytes in hospitalized COVID-19 patients. These increased levels are associated with disease severity [67,68]. These results corroborate the critical role of immune cell activation persistent in chronic kidney disease combined with the expression *ISG15*, *STAT1* and *MX1* could contribute to higher expression of pro-inflammatory cytokines potency and participate in the complication of SARS-CoV-2 infection.

Outstanding progress has been made in understanding the molecular basis of the antiviral actions of interferons. Moreover, studies have focused on delving into the strategies evolved by viruses to antagonize their actions [69,70,71]. Furthermore, advances in elucidating the IFN system have significantly contributed to our understanding of viral host patho-systems and disease outcomes. 

Basic and clinical research on related genes and drugs is required to provide more strategies for preventing and treating COVID-19 in DKD. Our knowledge of how molecular mechanisms underlying transcriptional and post-transcriptional events regulate the tunning in biological and metabolic pathways will contribute to translational research from the primary research laboratory to the clinic. According to our in-silico prediction, we proposed that the potential hub genes could have the following hierarchical order: *STAT1*, *MX1*, *ISG15*, *OAS1* and *IRF7,* as potential prognostic or therapeutic targets. However, further studies of validation in an extensive population, including other possible factors, such as smoking, obesity and geographical regions, will delineate more mechanisms underlying associations among multiple factors and COVID-19 that still need to be explored. We hope that the new knowledge generated in this study sheds light that could lead to developing novel therapeutic intervention strategies.

## 5. Conclusions

In summary, most differential expression analyses in DKD samples are upregulated, with ~97% of those identified to be related to induced COVID-19 pathways. Those genes participate in critical biological and metabolic processes, including complement and coagulation cascades, lipid and atherosclerosis, AGE-RAGE signaling pathway and positive regulation of cytokine production. Notably, those induced biomolecules include potential therapeutic targets for SARS-CoV-2 infection. Hence, transcriptional reprogramming of gene upregulation is a common mechanism in DKD and COVID-19 conditions are linked to possible pathways involved in the development of comorbidities. In addition, by performing interactomics analysis, we determine five hub genes co-expressed in modules participating in biological events associated with heart disease, virus immune response and genetic regulation at the transcriptional and post-transcriptional levels. Our expression profile analysis reveals that *ISG15* and *MX1* are the most conserved upregulated genes across COVID-19 datasets of immunological cells.

Meanwhile, *OAS1* and *MX1* are the genes with lower gene expression levels in tissue-specific healthy samples. Altogether, our results pointed out that these five hub genes could play an essential role in developing severe outcomes of COVID-19 in DKD patients. The results obtained in this bioinformatics analysis could contribute to establishing future strategies using key biomolecules for clinical decision-making in medical routine.

## Figures and Tables

**Figure 1 genes-13-02412-f001:**
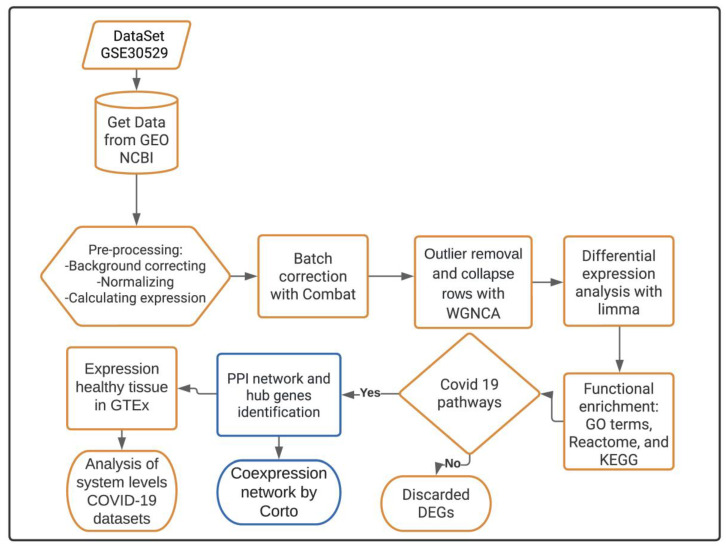
Bioinformatics pipeline for obtaining predictive hub genes involved in the pathophysiological impact of COVID-19 on DKD condition. Highlighted in yellow are the steps to perform the transcriptomics analysis, while those in blue are steps for the interactomics methodology. First, transcriptomic data were acquired from the NCBI GEO database. Data were pre-processed and outlier detection with WGCNA analysis was performed. Then, we obtained the DEGs and determined their potential functional roles in biological and molecular pathways. Those DEGs that overlapped among COVID-19 and DKD pathways were selected and interactome analysis was performed using protein–protein interaction (PPI) associations. Subsequently, the potential hub genes were identified using the Maximal Clique Centrality (MCC) topological algorithm. Then, their co-expressed gene modules were determined using the *Corto* algorithm. Finally, hub gene expression was validated using analysis of systems levels employing the Coronascape COVID-19 datasets and gene expression value comparisons in healthy tissues using the GTEx database.

**Figure 2 genes-13-02412-f002:**
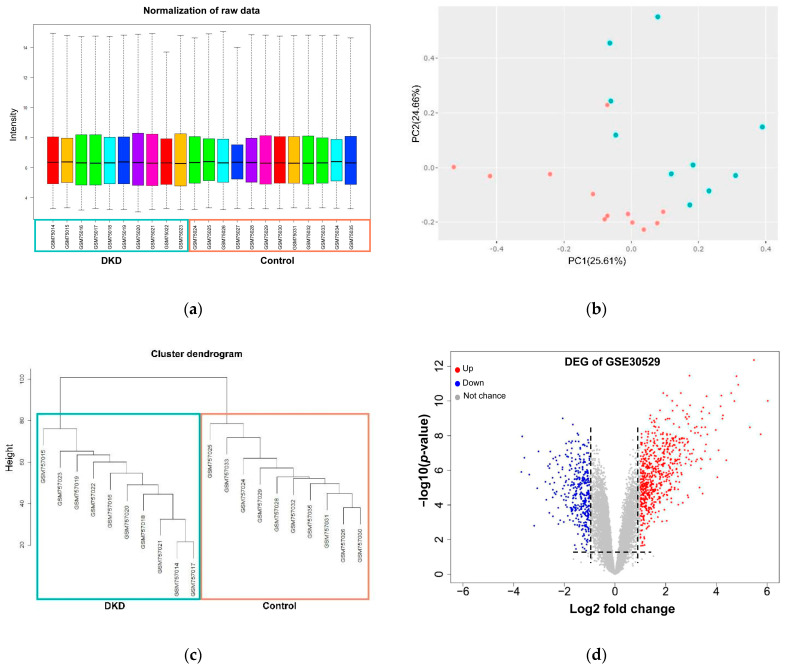
Pre-processing of GSE30529 dataset raw microarray data and differential expression analysis. (**a**) Boxplot of normalized raw data contrasting human patient healthy and DKD samples from kidney tubule. (**b**) PCA plots depict similarities and differences among DKD and control samples after data normalization. (**c**) WGCNA clustering represents 20 gene modules and two clusters. (**d**) Volcano plot of DEG, red (up) and blue (down) colored dots indicate the DEGs, while the grey dots represent genes without expression changes (NC) among DKD and control samples.

**Figure 3 genes-13-02412-f003:**
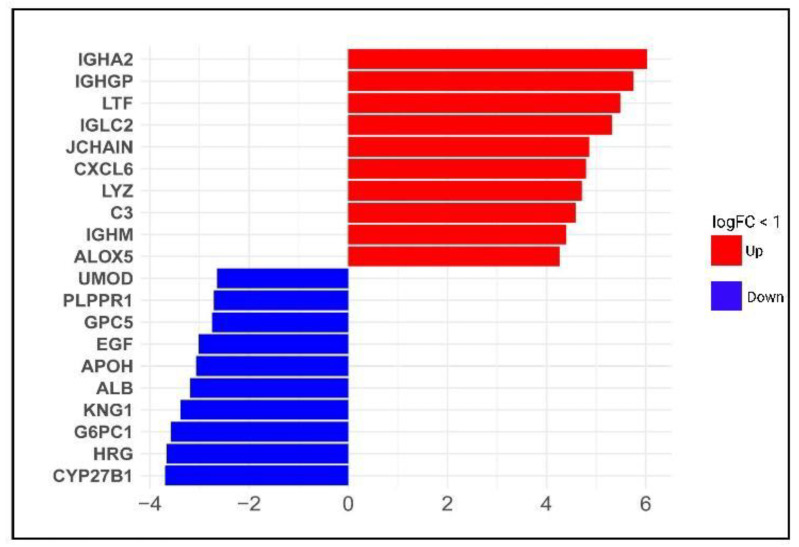
Top upregulated and downregulated DEGs in diabetic kidney disease dataset. Bars show the top upregulated (red) and downregulated (blue) genes, according to Log2FC values.

**Figure 4 genes-13-02412-f004:**
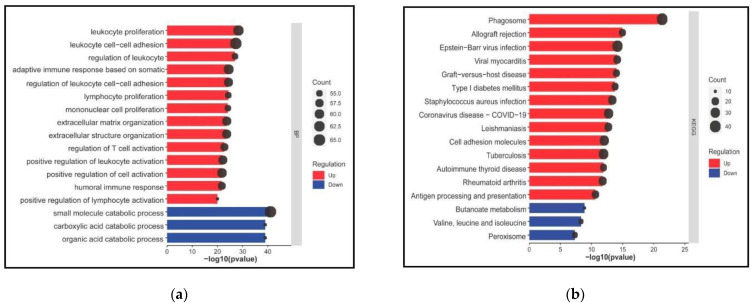
Functional enrichment analysis of DEGs. (**a**) Gene Ontology biological processes and (**b**) KEGG pathways. Bars for upregulated are colored in red while downregulated are depicted in blue.

**Figure 5 genes-13-02412-f005:**
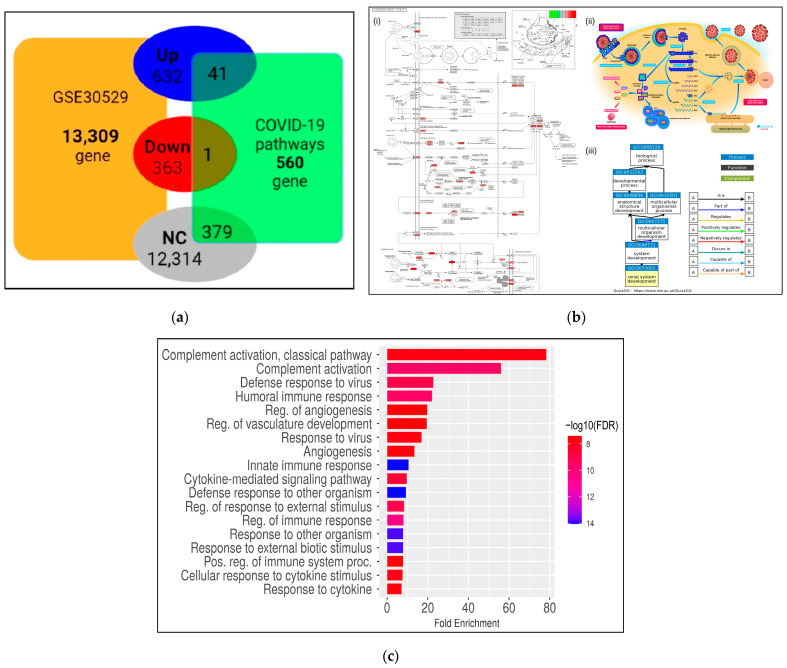
Identification of upregulated genes in DKD associated with COVID-19 pathways. (**a**) Diagram of overlapping DEGs among DKD and the COVID-19 pathways. The intersection of 41 genes in the COVID-19 pathway is shown. (**b**) (i) The KEGG pathway map hsa05171 is involved in Coronavirus disease-COVID-19. The rectangle in red color (light to dark), according to log Log2FC, indicates the upregulated DEGs overlapping DKD and COVID-19. (ii) the Reactome pathway R-hsa-9694516 and (iii) the GO term corresponding to renal system development. (**c**) Functional enrichment analysis of the shared genes among DKD and COVID-19 pathways. FDR is calculated based on a nominal *p*-value from the hypergeometric test. Fold Enrichment is defined as the percentage of DEGs belonging to a pathway divided by the corresponding percentage in the background. FDR reports how likely the Enrichment is by chance. Higher values are colored on a scale of red to blue. In the *x*-axis, Fold Enrichment indicates how drastically genes of a specific pathway are overrepresented.

**Figure 6 genes-13-02412-f006:**
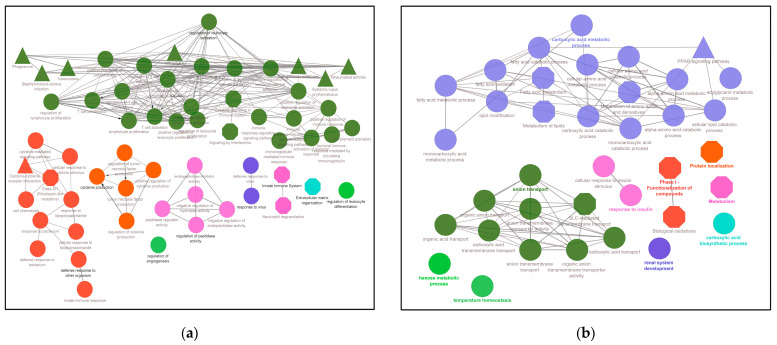
Network analysis of the functional role of deregulated genes in diabetic kidney disease. (**a**) Inter-relational pathway enrichment analysis is shown for the upregulated DEGs in DKD samples; (**b**) for the downregulated DEGs in DKD; (**c**) for the 41 DEGs overlapping DKD and COVID-19 pathways. Circle represents GO terms, triangle and octagon represent KEGG and REACTOME, respectively, while green, orange, purple and pink color represent pathways.

**Figure 7 genes-13-02412-f007:**
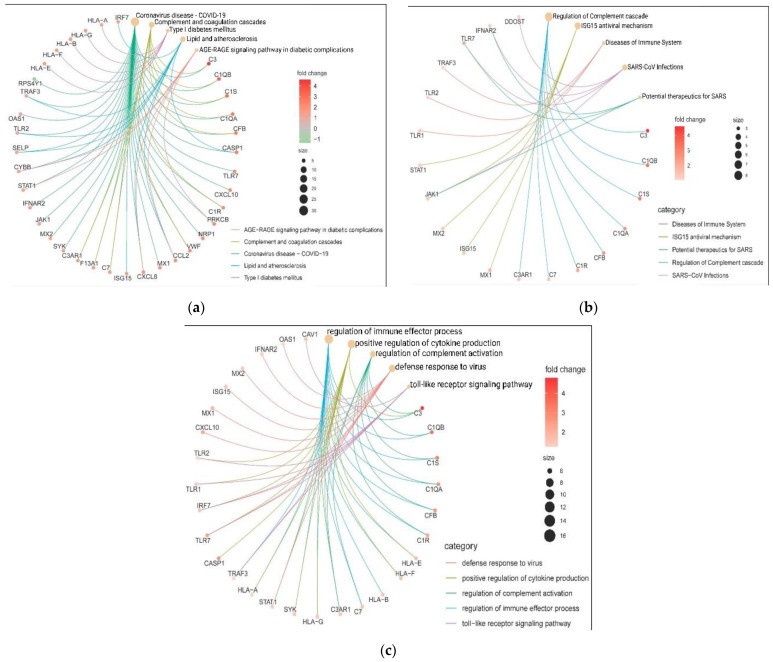
Category cnetplot depicts the linkages of upregulated genes and the biological pathways as a network. Cnetplot of (**a**) KEGG, (**b**) Reactome, and (**c**) Biological process shows the pathways and associated 41 up-DEGs networks. Red (light to dark) nodes indicate the Log2FC values, while yellow nodes indicate the functional enriched pathways whose size represents the number of genes that are DEGs in each of them.

**Figure 8 genes-13-02412-f008:**
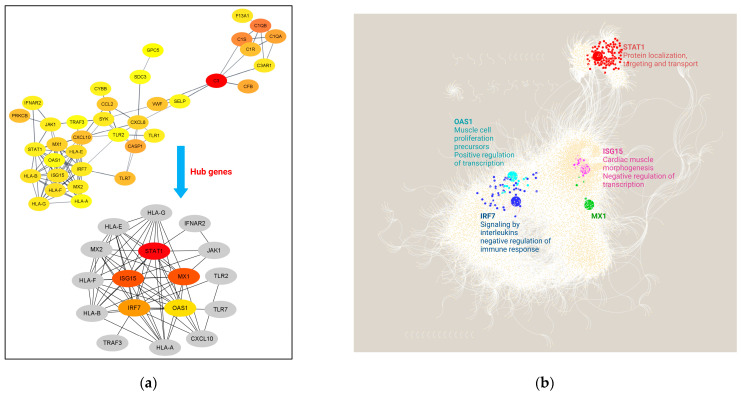
Interactome analysis of shared DEGs among DKD and COVID-10 pathways. (**a**) PPI network of DEGs and their closest interactors. Edges represent experimental validated interactions among proteins with a high score value (0.7). Hub genes were retrieved from cytoHubba and are shown according to score nodes (from yellow to red); (**b**) Interactome of the five identified hubs genes. Each node represents a gene and the edges between nodes represent regulatory interactions among genes. The big, highlighted nodes with specific color represent each of the hub genes, while nodes sharing the same color surrounding these represent co-expressed gene modules. Enriched biological processes of each module are depicted in the same color as well. *STAT1* (red); *OAS1* (turquoise); *ISG15* (pink); *IRF7* (navy blue); *MX1* (green).

**Figure 9 genes-13-02412-f009:**
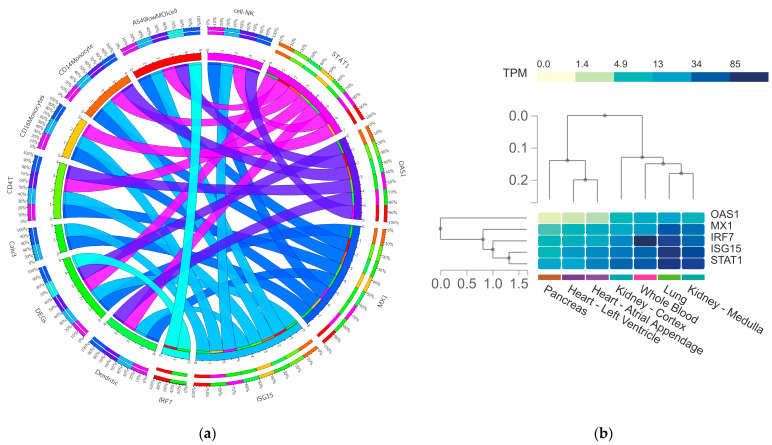
In-silico validation of gene expression of hub genes across other databases. (**a**) Circa plot depicts the number of shared DEGS across the 8 types of COVID-19 datasets under study. In the hemisphere, the 5 hub DEGs are represented by an arc: *OAS1*, *STAT1*, *CXCL10*, *ISG15* and *MX1*. The inner arcs represent the distribution of the hub genes in each COVID-19 dataset, while the outside arcs represent the intersection of both hemispheres on a scale of 0–100%. Data was obtained from the Coronascape database including dendritic cells, Calu3, CD4T, CD16 monocytes, CD14 monocytes, NK cells and A549lowMOI cell samples (https://metascape.org/COVID/, accessed on 14 September 2022). (**b**) Heatmap comparison of hub ‘ healthy tissue expression and (**c**) lung single-cell expression using the GTEx DB (https://www.gtexportal.org, accessed on 16 September 2022).

**Figure 10 genes-13-02412-f010:**
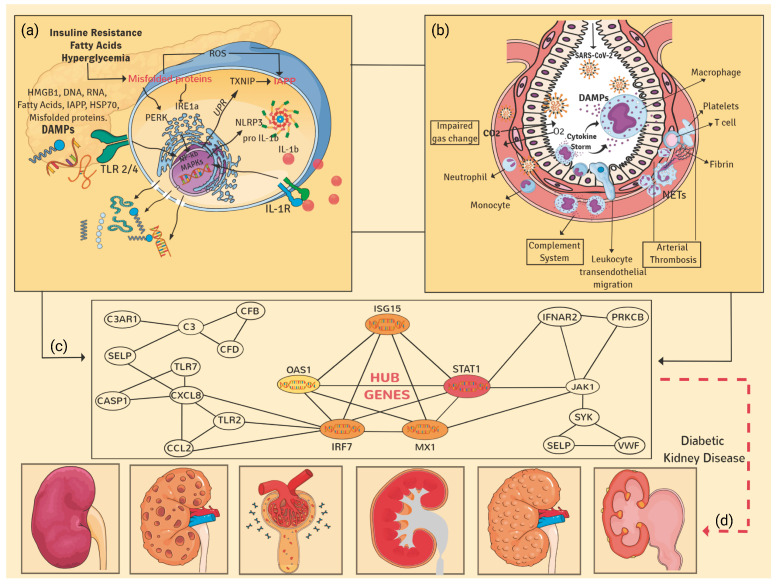
Pathophysiological impact of COVID-19 on DKD condition. (**a**) An illustration that simplifies the multiple processes and proinflammatory pathways that initiate in an orchestrated manner, the primary stage of the chronic inflammatory response in pancreatic cells coupled with the activation of the complement system due to the presence of DAMPs, excessive synthesis of ROS and dysfunction endothelial, contribute to the progression of DM and the development of micro- and macrovascular complications in target organs, (**b**) pathogen-host interaction mechanisms, showing the alveolar microenvironment, where the localized inflammatory response is initiated by the presence of PAMPs and DAMPs, with the consequent endothelial activation, migration of immune cells, significant tissue damage and uncontrolled release of cytokines, as well as proinflammatory interleukins (cytokine storm). (**c**) Predictive hub genes and their impact on the pathophysiological process of diabetic kidney disease and SARS-CoV-2 infection; (**d**) representation of the multiple functional and morphological alterations in renal tissue due to dysfunction of podocytes, epithelial cells, endothelial cells and local macrophages, events that ultimately triggers the development of glomerular sclerosis, hyalinosis, deposition in the mesangial extracellular matrix and local fibrosis, alterations which together increased renal filtration, precipitation of glomerular nephrosis, acute kidney injury and progressive chronic kidney disease (CKD), among others comorbidities.

**Table 1 genes-13-02412-t001:** Top 15 DEGs overlapped among DKD and COVID-19 pathways.

Gene Description	Gene Symbol	KEGG	Reactome
Complement C3	C3	*	NA
Caspase 1	CASP1	*	NA
C-C motif chemokine ligand 2	CCL2	*	NA
Complement factor B	CFB	*	*
C-X-C motif chemokine ligand 10	CXCL10	*	NA
C-X-C motif chemokine ligand 8	CXCL8	*	NA
Interferon α and β receptor subunit 2	IFNAR2	*	*
Interferon regulatory factor 7	IRF7	*	*
ISG15 ubiquitin like modifier	ISG15	*	*
Janus kinase 1	JAK1	*	*
MX dynamin like GTPase 1	MX1	*	NA
2′-5′-oligoadenylate synthetase 1	OAS1	*	*
Signal transducer and activator of transcription 1	STAT1	*	*
Toll like receptor 2	TLR2	*	NA
Toll like receptor 7	TLR7	*	NA

Presence: (*) or not (NA).

## Data Availability

Datasets used are available from GEO repository (https://www.ncbi.nlm.nih.gov/gds) (accessed on 22 June 2022). Code and Appendix A have been deposited in GitHub repository.

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
