# Peer review of "In Silico Prediction of Hub Genes Involved in Diabetic Kidney and COVID-19 Related Disease by Differential Gene Expression and Interactome Analysis"

_genes, 2022, doi:10.3390/genes13122412_

Round 1
Reviewer 1 Report
This is a very well written manuscript.
Figure 1 text is fuzzy, clean up presentation of text.
Page 7, didn't mention UMOD in the text among the top 10 down regulated
Page 8, line 292, find a better reference than just genecards.org
Page 11, line 392, find a better reference than just genecards.org for OAS1
Author Response
We thank reviewer #1 for their relevant and valuable comments. In this document, we quote statements from the reports in boldface, and our responses follow in ordinary print. The corresponding modifications and corrections made in the revised manuscript are summarized in our response below.

Reviewer 2 Report
The authors summarize that most differential expression analysis in DKD samples are upregulated, with ~ 97% of those identified to be related to COVID-19 pathways induced. Those genes participate in pivotal biological and metabolic processes, including complement and coagulation cascades, lipid and atherosclerosis, AGE-RAGE signaling pathway, and positive regulation of cytokine production. Notably, those induced biomolecules include potential therapeutic targets for SARS-CoV-2 infection.
Introduction
1. The introduction part could be streamlined, especially paragraph 5 and 6.
Materials and Methods:
1. In the content of https://github.com/kap8416/Transcriptomics-Diabetes-Kidney-Disease.; the authors describe “ human samples from patients with DKD (n = 10) and healthy (non-diabetic) controls”. How to define the sample numbers of the cases (e.g. instead of n=15?) Do all DKD cases and healthy controls have previously confirmed COVID-19 infection?
2. How many kinds of experimental tissue samples collected after COVID-19 infection in both healthy and DKD cases (e.g. kidney, lung) ?
3. Did the authors also exam the DEG between non-DM CKD and healthy individuals after COVID-19 infection?
Author Response
We thank reviewer #2 for their relevant and valuable comments. The English language was significantly improved in the manuscript. After an exhaustive revision, we considered it acceptable for publication. In this document, we quote statements from the reports in boldface, and our responses follow in ordinary print. The corresponding modifications and corrections made in the revised manuscript are summarized in our response below.
